# Prediction Model with *HLA-A*33:03* Reveals Number of Days to Develop Liver Cancer from Blood Test

**DOI:** 10.3390/ijms24054761

**Published:** 2023-03-01

**Authors:** Nao Nishida, Jun Ohashi, Goki Suda, Takehiro Chiyoda, Nobuharu Tamaki, Takahiro Tomiyama, Sachiko Ogasawara, Masaya Sugiyama, Yosuke Kawai, Seik-Soon Khor, Masao Nagasaki, Akihiro Fujimoto, Takayo Tsuchiura, Miyuki Ishikawa, Koichi Matsuda, Hirohisa Yano, Tomoharu Yoshizumi, Namiki Izumi, Kiyoshi Hasegawa, Naoya Sakamoto, Masashi Mizokami, Katsushi Tokunaga

**Affiliations:** 1Genome Medical Science Project, National Center for Global Health and Medicine, Ichikawa 272-8516, Japan; 2Department of Genomic Function and Diversity, Medical Research Institute, Tokyo Medical and Dental University, Tokyo 113-8510, Japan; 3Department of Biological Sciences, Graduate School of Science, The University of Tokyo, Tokyo 113-0033, Japan; 4Department of Gastroenterology and Hepatology, Graduate School of Medicine, Hokkaido University, North 15, West 7, Kita-ku, Sapporo 060-8638, Japan; 5Hepato-Biliary-Pancreatic Division, Department of Surgery, Graduate School of Medicine, The University of Tokyo, Tokyo 113-8655, Japan; 6Department of Gastroenterology and Hepatology, Musashino Red Cross Hospital, Musashino 180-8610, Japan; 7Department of Surgery and Science, Graduate School of Medical Sciences, Kyushu University, Fukuoka 812-8582, Japan; 8Department of Pathology, Kurume University School of Medicine, Kurume 830-0011, Japan; 9Department of Viral Pathogenesis and Controls, National Center for Global Health and Medicine, Ichikawa 272-8516, Japan; 10Genome Medical Science Project-Toyama, National Center for Global Health and Medicine, Tokyo 162-8655, Japan; 11Human Biosciences Unit for the Top Global Course Center for the Promotion of Interdisciplinary Education and Research, Kyoto University, Kyoto 606-8507, Japan; 12Department of Human Genetics, Graduate School of Medicine, The University of Tokyo, Tokyo 113-0003, Japan; 13Laboratory of Clinical Genome Sequencing, Department of Computational Biology and Medical Sciences, Graduate School of Frontier Sciences, The University of Tokyo, Tokyo 108-8639, Japan

**Keywords:** hepatitis B virus, hepatocellular carcinoma, human leukocyte antigen, prediction, cox proportional-hazards regression model

## Abstract

The development of liver cancer in patients with hepatitis B is a major problem, and several models have been reported to predict the development of liver cancer. However, no predictive model involving human genetic factors has been reported to date. For the items incorporated in the prediction model reported so far, we selected items that were significant in predicting liver carcinogenesis in Japanese patients with hepatitis B and constructed a prediction model of liver carcinogenesis by the Cox proportional hazard model with the addition of *Human Leukocyte Antigen* (*HLA*) genotypes. The model, which included four items—sex, age at the time of examination, alpha-fetoprotein level (log_10_AFP) and presence or absence of *HLA-A*33:03*—revealed an area under the receiver operating characteristic curve (AUROC) of 0.862 for HCC prediction within 1 year and an AUROC of 0.863 within 3 years. A 1000 repeated validation test resulted in a C-index of 0.75 or higher, or sensitivity of 0.70 or higher, indicating that this predictive model can distinguish those at high risk of developing liver cancer within a few years with high accuracy. The prediction model constructed in this study, which can distinguish between chronic hepatitis B patients who develop hepatocellular carcinoma (HCC) early and those who develop HCC late or not, is clinically meaningful.

## 1. Introduction

About 350 million people are estimated to be infected with the hepatitis B virus (HBV) worldwide, and about 1.3 million to 1.5 million people are infected with the virus in Japan. Although the Japanese government started a universal hepatitis B vaccination program from October 2016, it is estimated that there are currently about 10,000 new cases annually. In a multipurpose Japanese cohort study, it was reported that patients with HBV infection had a 16.1-fold increased risk of liver carcinogenesis [1], and there is a strong demand for accurate prediction of carcinogenesis in HBV-infected individuals.

Several clinical parameters have been reported as risk factors for liver carcinogenesis in HBV-infected patients, including HBV-DNA levels, presence of hepatitis B e-antigen (HBeAg), and presence of cirrhosis, in addition to basic information such as age and sex. REACH-B, constructed in a cohort study in Taiwan, is one of the most commonly used hepatocellular carcinoma (HCC) prediction models for untreated hepatitis B patients, with sex, age, serum alanine aminotransferase (ALT) level, HBeAg, and serum HBV-DNA levels used as variables [2]. REACH-B used Cox multivariate proportional hazards model to predict risk of HCC at 3, 5, and 10 years, with an area under the receiver operating characteristic curve (AUROC) of 0.8111 at 3 years, 0.796 at 5 years, and 0.769 at 10 years. The PAGE-B is another commonly used the HCC predictive model for treated hepatitis B patients in Caucasians, using age and sex and platelet as variables [3]. The 5-years cumulative HCC incidence rates predicted by the PAGE-B model showed 0.82 of c-index after bootstrap validation for 1325 adult Caucasians who received entecavir/tenofovir for 12 months or longer. Subsequently, a modified PAGE-B (mPAGE-B) model was constructed by adding serum albumin level to the PAGE-B model in 2001, treated Asian hepatitis B patients, and showed 0.82 of AUROC [4].

In addition to REACH-B, the HCC predictive model for chronic hepatitis B (CHB) patients who remained untreated or were treated with entecavir only or with entecavir and tenofovir disoproxil fumarate includes GAG-HCC, reported by a Hong Kong group [5]. A risk score using the GAG-HCC model, which was based on age, gender, HBV-DNA levels, HBV core promoter mutations, and liver cirrhosis (LC), can estimate the development of HCC in 5- and 10-years with AUROC of 0.88 at 5 years, and 0.89 at 10 years. On the other hand, in addition to PAGE-B, the HCC predictive model for treated hepatitis B patients includes APA-B, reported by a Taiwanese group [6]. APA-B was the HCC prediction model based on age, sex, platelet counts, and alpha-fetoprotein (AFP) levels, and the prediction accuracy of HCC risk after 2, 3, and 5 years was 0.877, 0.842, and 0.827, respectively.

Several models have been reported to predict the risk of HCC in CHB patients [7], but none has added host genetic factors. Genome-wide association studies (GWASs) identified the association of *HLA* class II genes with chronic hepatitis B infection in Asian populations [8,9]. As a result of GWAS and *HLA* association analysis focusing on HCC in Japanese CHB patients, *HLA-DPB1*02:01*, which belongs to *HLA* class II, and *HLA-A*33:03* and *HLA-A*31:01*, which belongs to *HLA* class I, showed a significant association with HCC [10,11]. In this study, we constructed a model that added *HLA* genotypes to the prediction of HCC in hepatitis B patients and verified its predictive accuracy.

## 2. Results

### 2.1. Clinical Characteristics of Studied Individuals and Study Design

Clinical information from 893 hepatitis B patients distributed by the BioBank Japan (BBJ) was used to select clinical items that were significantly associated with the prediction of HCC [12,13]. Nine test items (including age, sex, HBV-DNA, HBeAg, ALT, LC, PLT, Alb, and AFP) incorporated into five previously reported HCC prediction models (REACH-B, GAG-HCC, PAGE-B, mPAGE-B, and APA-B) were examined to be associated with the development of HCC using BBJ samples. A stepwise variable selection based on the Akaike information criterion (AIC) was performed for the Cox proportional-hazards regression model with the nine variables. As a result, four items were selected: presence or absence of LC, log_10_AFP, age, and sex. An independent set of 672 hepatitis B patients were collected from four hospitals and determined their HLA genotypes. In Table 1, clinical information of 162 patients which were able to collect four clinical items without any defects was summarized, along with the presence or absence of HLA-A*33:03, HLA-A*31:01, and HLA-DPB1*02:01. Clinical information of all 672 patients, including missing values, is summarized in Materials and Methods.

For all 162 cases, the number of days between the date of the blood test and the date of the liver cancer diagnosis, or between the date of the blood test and the date of the investigation, is available (Figure 1). Non-HCC subjects in which 5 years had not elapsed between the date of the blood test and the date of the investigation were excluded from the analysis.

### 2.2. The Cox Proportional-Hazards Regression Model to Predict the Early Onset of HCC

To predict which CHB patients would develop HCC after blood test, we used the Cox proportional-hazards regression model [14]. The results of multivariate Cox proportional-hazards regression analysis for seven variables (age, sex, LC, log_10_AFP, *HLA-A*31:01*, *HLA-A*33:03*, and *HLA-DPB1*02:01*) are described in Materials and Methods. After the stepwise variable selection using 162 cases with no lack of clinical information, four variables remained: age, sex, log_10_AFP, and *HLA-A*33:03* (Table 2). The hazard function is *h*(*t*) = *h*_0_(*t*)*exp(0.057*age + 0.601*sex (male = 1, female = 0) + 0.642*log_10_AFP + 0.714**A*33:03* (carrier = 1, non-carrier = 0)), where *h*_0_(*t*) indicates the baseline hazard function. As an example, the beta coefficient for age is 0.057, indicating that older age is associated with a higher risk of liver cancer. Table 2 shows beta coefficient, hazard ratio (HR), confidence intervals of the hazard ratios (HR lower and HR upper), and global statistical significance of the model (z and *p*.value). From a clinical standpoint, it is important to distinguish between CHB patients who develop HCC early and those who develop HCC late or not. The value of *h*_0_(*t*) varies with time but is common to all the subjects. Therefore, to facilitate the use of the results of this study, score noted by *S* was calculated for each subject according to the following formula:(1)$$\begin{array}{c} S=0.057\times \mathrm{age}+0.601\times \mathrm{sex}\;(\mathrm{male}=1,\mathrm{female}=0)+0.642\times \log_{10}\mathrm{AFP}+\\ 0.714\times A*33:03\;(\mathrm{carrier}=1,\mathrm{carrier}=0) \end{array}$$

The distribution of *S* is shown in Figure 2. We then used receiver operating characteristic (ROC) analysis to determine whether *S* could be used to predict the early onset of HCC. The study design of ROC analysis for HCC onset within X years was summarized in Figure 3. The area under the curve (AUC) was more than 0.85 when discriminating between patients who would develop HCC within 1 year and those who would develop HCC later or not (Figure 4a). For the case of within 3 years, the sensitivity exceeded 0.8 (Figure 4b). The results from ROC analyses at various numbers of years suggest that the performance is better at discriminating patients who would develop HCC within a few years (Figure 4c). The survival curves for subjects with *S* of more than 4 and with *S* of less than or equal to 4 are shown in Figure 5. *S* values more than 4 were associated with a higher probability of developing HCC within 100 days. From the present study, we can say that a score above 4 indicates a high likelihood of developing HCC within a few years.

### 2.3. Validation of the Model

Since the samples used to construct the Cox proportional-hazards regression model was also used in the ROC analysis, the performance mentioned above is overestimated. Thus, to evaluate the performance of the Cox proportional-hazards regression model with four variables, computer simulations were further performed. In the simulation, 162 subjects were first randomly shuffled and then split into a pair of training and test sets without considering whether they were CHB or HCC. The results form training set were used to predict the outcome of each subject in test set. After 1000 repetitions of this operation, the distributions of Harrell’s C-index [15], sensitivity, and specificity were obtained (Figure 6). The model constructed in this study appears to perform well when detecting patients who would develop HCC within a year, since the mean of Harrell’s C-index and the mean of sensitivity exceeded 0.75 and 0.7, respectively.

## 3. Discussion

In order to predict the risk of liver carcinogenesis in Japanese patients with hepatitis B, we constructed a prediction model by adding *HLA* to items that had been incorporated into previous HBV related HCC prediction models, including basic information (including sex, year of birth, and age at first year registration) and blood test results (including HBV-DNA levels, presence or absence of HBe antigen, serum ALT level, platelet counts, serum albumin level, and AFP level). Among these items, the BBJ sample was used to examine four items showing significant association with predicting liver carcinogenesis in Japanese patients with hepatitis B: age, sex, log_10_AFP, and presence of liver cirrhosis. Among 672 Japanese patients with hepatitis B who were independent of the BBJ sample, we constructed a prediction model for liver carcinogenesis by adding *HLA* genotypes in 162 cases without any deficiency in the four items. We used the Cox proportional-hazards regression model to predict which CHB patients would develop HCC after blood test, and, as a result, we completed the prediction model based on the four items of age, sex, log_10_AFP and *HLA-A*33:03*. Of the four items included in the HCC prediction model constructed in this study, age was included in all five previously reported prediction models. Sex is also an item incorporated into four of the five prediction models (REACH-B, GAG-HCC, PAGE-B, and mPAGE-B). AFP is an item included only in APA-B among the five prediction models, but it is also included in other HCC prediction models, REAL-B [16] and RWS-HCC [17]. An HCC prediction model incorporating the *HLA* genotype has never been reported before, and the HCC prediction model constructed in this study is the first report. For AUROC for 3-year HCC prediction, the prediction model constructed in this study had AUROC = 0.86, which was second only to AUROC = 0.88 for GAG-HCC, which had the highest AUROC among the five previously reported prediction models.

To validate the performance of the HCC prediction model based on four variables, 162 cases were randomly shuffled 2:1 as a training set and a test set, and the accuracy was evaluated by predicting the disease state of the test set with the training set. After 1000 repeated evaluations, A C-index of 0.75 or higher and a sensitivity of 0.70 or higher indicated that the model can accurately detect patients at high risk of developing liver cancer. As a result of the AUC from the ROC analysis, the AUC was higher for years closer to the date of the blood test, indicating that the HCC prediction model can predict the development of early liver cancer with high accuracy. Although sensitivity at 1 year after blood test fell below 0.7, both sensitivity and specificity tended to decrease over time, as did AUC. We also found that patients with a predicted score “S” of 4 or higher had a significantly increased risk of developing liver cancer within 100 days. The prediction model constructed in this study, which can distinguish between CHB patients who develop HCC early and those who develop HCC late or not, is clinically meaningful.

Although it is not a prediction model that reflects the incidence of liver cancer in the natural population because the cases were collected retrospectively, it is possible to predict the period from the date of blood test to the time of liver cancer. Prediction of the time to liver cancer based on the condition of the blood test date is expected to provide clinically useful information because appropriate treatment and medication can be given to patients. However, because *HLA* is known to have very different frequencies of genotypes among populations, the limitation of this study is that the HCC prediction model including *HLA* genotypes constructed in this study may be less effective in non-Asians. In the future, it is necessary to validate the model with prospectively collected cases, and to test the efficacy in non-Asian populations.

The conclusion of this study is that the current model that can predict the development of liver cancer within a year with high accuracy in hepatitis B patients who have specific *HLA* genotype can be expected to be a clinically useful tool.

## 4. Materials and Methods

### 4.1. Study Population

A total of 672 Japanese patients aged 18 years and older with chronic HBV infection were recruited in four hospitals, including Hokkaido University Hospital, the University of Tokyo Hospital, Kyushu University Hospital, and Kurume University Hospital. Basic information (including age and sex), blood test results (including date of test, HBV-DNA levels, presence or absence of HBe antigen, ALT, PLT, Alb, and AFP), and date of diagnosis of HCC for each patient were collected from each hospital using the same inquiry form (Table 3). Written informed consent was obtained from all patients involved in the study.

In addition, an independent set of 893 Japanese HBV patients aged 18 years and older was distributed by the BBJ and was used to select the clinical test items to be used in the construction of the HCC prediction model. A total of more than 300 items of clinical information distributed by the BBJ include basic information (including sex, year of birth, and age at first year registration) and blood test results (including HBV-DNA levels, presence or absence of HBe antigen, serum ALT level, platelet counts, serum albumin level, and AFP level). The use of distributed data from the BBJ for this study has been approved by both the NCGM and BBJ ethics committees.

### 4.2. Data Curation of Clinical Items

The outlier test by Smirnov–Grubbs test was performed for the continuous clinical value. Although there were several cases of ALT exceeding 1000 U/L, we decided not to perform the outlier test because patients with acute hepatitis may have elevated ALT levels. There are cases in which AFP exceeds 1000 ng/mL, but we decided not to perform the outlier test because AFP is higher in patients who develop HCC. For PLT with a normal value in the range of 100,000/μL to 454,000/μL and Alb with a normal value of 3.5 g/dL or more, outliers were determined to be excluded by Smirnov–Grubbs test.

### 4.3. HLA Imputation Using Genome-Wide SNP Typing Data

For 511 of the total 672 cases, genome-wide SNP typing was performed on genomic DNA extracted from blood samples collected from patients, and HLA genotype were determined by HLA imputation. Genome-wide SNP typing was carried out using the AXIOM Genome-Wide ASI 1 array, according to the manufacturer’s instructions. SNP data from an extended MHC region ranging from 25,759,242 to 33,534,827 bp based on the hg19 position was used to conduct 2-field HLA genotype imputation for six class I and class II *HLA* genes using the HIBAG R package the same way as in our previous report [18]. The remaining 161 cases were subjected to whole-genome sequencing using DNA extracted from non-cancerous portions of hepatocellular carcinoma tissue samples. Whole genome sequencing (WGS) was performed using the TruSeq DNA PCR Free Kit (Illumina) to create libraries and NovaSeq 6000 (Illumina) to acquire data at 150 bp pair-end, according to the manufacturer’s instructions. *HLA* genotypes to WGS were determined by *HLA* calling using HLA-VBSeq pipeline [19,20]. Initially, a preprocessing pipeline aligned the human reference assembly GRCh38 without alt contigs and reference sequence of HBV genotypes A to D (GencodeID: AB246337, AB246339, AB246344 and AF043594) using bwa (ver. 0.7.17) [21]. Next a preprocessing step extracted the sequenced reads mapped to the *HLA* chromosomal regions and unmapped sequenced reads. Using these extracted reads and the *HLA* reference panel constructed from the IPD-IMGT3.45.0 reference sequences [22], the HLA-VBSeq pipeline [20] called the *HLA* genotype combinations.

### 4.4. Constructing the Cox Proportional-Hazards Regression Model to Predict the Onset of HCC

In this study, to construct the Cox proportional-hazards regression model for predicting the onset of HCC in CHB patients, we used 162 subjects (83 HCC and 79 CHB patients) with no missing values for the following seven variables: age, sex, LC, log_10_AFP, *HLA-A*33:03*, *HLA-A*31:01*, and *HLA-DPB1*02:01*. A stepwise variable selection based on the AIC was performed for the Cox proportional-hazards regression model with the seven initial variables (Table 4). The survival time of each subject was defined as the time from blood test date to date of liver cancer onset (when CHB patient developed HCC) or to the date of censoring (when CHB patient did not develop HCC at the time of study). A receiver operating characteristic (ROC) analysis was conducted to examine if the hazard function obtained from the Cox proportional-hazards regression model with four variables—age, sex, log_10_AFP, and *HLA-A*33:03*—could be used to predict CHB patients who develop HCC within the specified number of years. In the ROC analysis, CHB patients who developed HCC later than the specified number of years were regarded as ones who did not develop HCC, and CHB patients who were censored within the specified number of years were excluded from the analysis. The optimal threshold was defined as the point closest to the top-left corner in the ROC plane. Sensitivity and specificity were calculated at the threshold. The R packages “survival” and “pROC” were used to carry out the Cox proportional-hazards regression analysis and the ROC analysis, respectively. The survival curve analysis was conducted using R package “survminer”.

### 4.5. Validation of the HBV-HCC Model by Computer Simulation

To evaluate the performance of the Cox proportional-hazards regression model with four variables, computer simulations were performed. A total of 162 subjects were first randomly shuffled and then split into a pair of training and test sets without considering whether they were CHB or HCC (training and test sets were composed of 2/3 and 1/3 of the 162 subjects). The training set was used to construct a Cox proportional-hazards regression model, and the threshold in the ROC analysis for the specified number of years was obtained. The results of the training set were used to compute the hazard function for each subject in the test set, and Harrell’s C-index, sensitivity, and specificity were obtained for the test set. This procedure was repeated 1000 times.

## Figures and Tables

**Figure 1 ijms-24-04761-f001:**
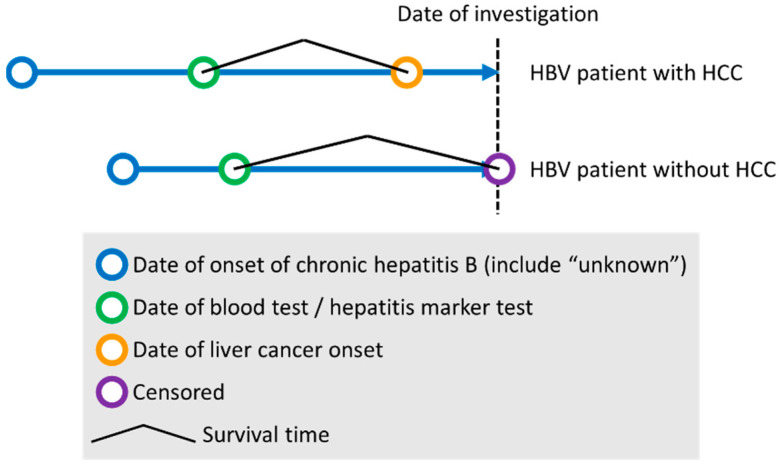
Study design for Cox proportional hazards regression analysis.

**Figure 2 ijms-24-04761-f002:**
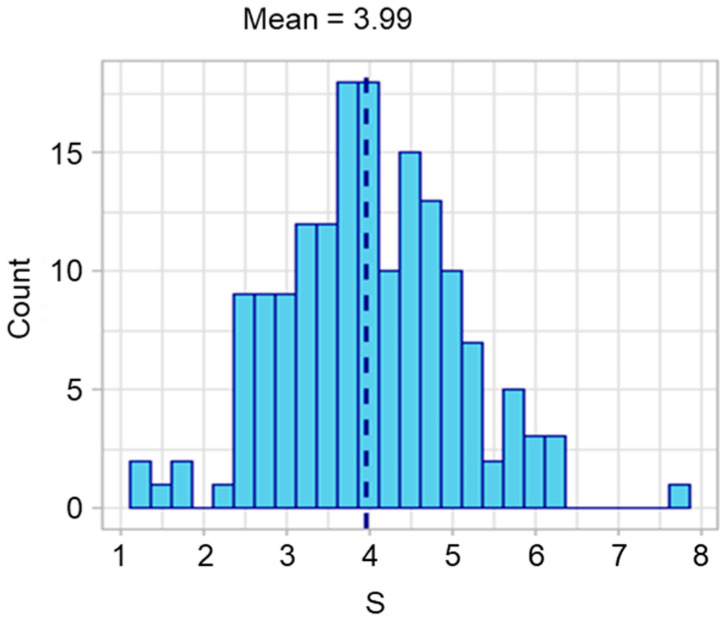
Distribution of S in 162 subjects. A dashed line indicates the mean of S in 162 cases with no lack of clinical information.

**Figure 3 ijms-24-04761-f003:**
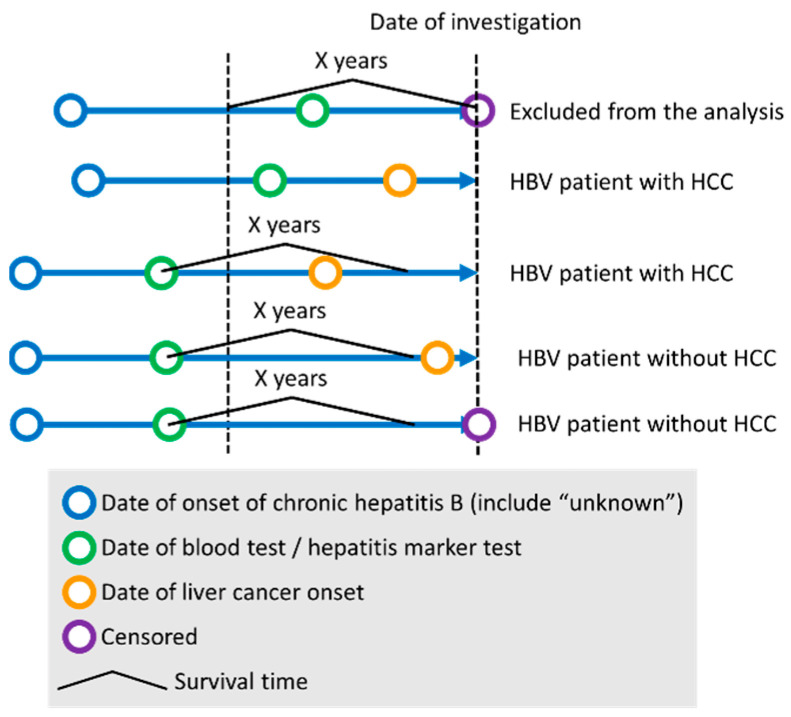
Study design of ROC analysis for HCC onset within X years.

**Figure 4 ijms-24-04761-f004:**
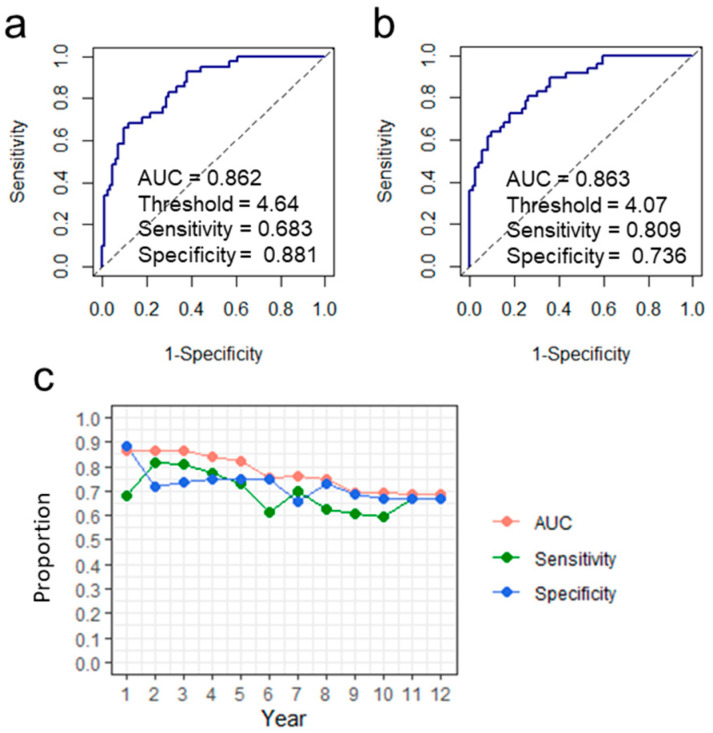
Results from the ROC analysis. ROC for the case of discriminating between patients who would develop HCC within 1 year and those who would develop HCC later or not (**a**). ROC for the case of discriminating between patients who would develop HCC within 3 years and those who would develop HCC later or not (**b**). AUC, sensitivity, and specificity obtained from the ROC analysis for the case of discriminating between patients who would develop HCC within the specified number of years and those who would develop HCC later or not (**c**).

**Figure 5 ijms-24-04761-f005:**
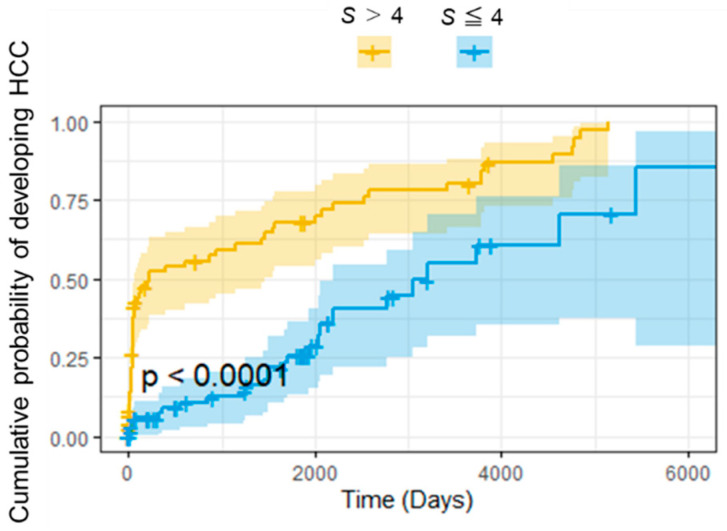
Survival curves for subjects with S of more than 4 and with S of less than or equal to 4. *p* value in the plot was calculated by log-rank test.

**Figure 6 ijms-24-04761-f006:**
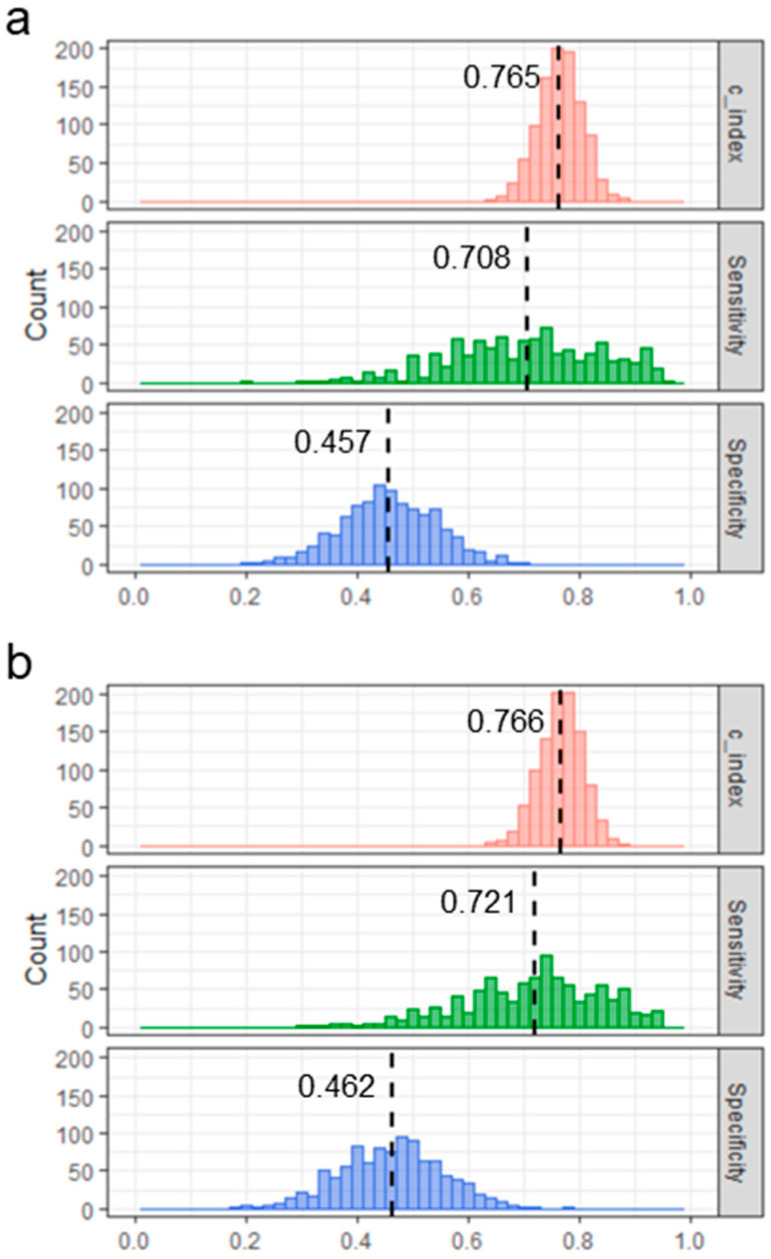
Harrell’s C-index, sensitivity, and specificity obtained from computer simulation. The case of discriminating between patients who would develop HCC within 1 year and those who would develop HCC later or not (**a**). The case of discriminating between patients who would develop HCC within 3 years and those who would develop HCC later or not (**b**).

**Table 1 ijms-24-04761-t001:** Clinical information of 162 HBV patients without defects in 7 items.

	HBV Patients with HCC (*n* = 83)	HBV Patients without HCC (*n* = 79)
Age at blood test (average (min, max))	57.1 (32–83)	48.7 (21–78)
Average number of days from blood test to diagnosis of HCC	1274	-
Average number of days between blood test date and censored date	-	1189
Sex (Male/Female)	65/18	34/45
Presence of LC at blood test (%)	12.0	5.1
AFP at blood test (median log_10_AFP)	1.16	0.55
*HLA-A*33:03* carrier frequency	0.11	0.08
*HLA-A*31:01* carrier frequency	0.24	0.28
*HLA-DPB1*02:01* carrier frequency	0.33	0.38

**Table 2 ijms-24-04761-t002:** Cox proportional hazards model with four associated variables.

Variable	Beta	HR	HR Lower	HR Upper	z	*p* Value
age	0.057	1.058	1.036	1.081	5.313	1.08 × 10^−7^
sex	0.601	1.825	1.069	3.114	2.206	2.74 × 10^−2^
log_10_AFP	0.642	1.900	1.467	2.462	4.859	1.18 × 10^−6^
*A*33:03*	0.714	2.042	0.987	4.225	1.924	5.44 × 10^−2^

HR: hazard ratio.

**Table 3 ijms-24-04761-t003:** Clinical information of all studied samples.

	HBV Patients with HCC (*n* = 291)	HBV Patients without HCC (*n* = 361)
Age at blood test (average (min, max))	58.4 (37–83)	53.9 (18–82)
Average number of days from blood test to diagnosis of HCC	1130	-
Average number of days between blood test date and censored date	-	2826
Sex (Male/Female)	233/58	169/192
Presence of LC at blood test (%)	26.8	4.71
AFP at blood test (median log_10_AFP)	1.33	0.57
*HLA-A*33:03* carrier frequency	0.10	0.08
*HLA-A*31:01* carrier frequency	0.228	0.20
*HLA-DPB1*02:01* carrier frequency	0.398	0.336

**Table 4 ijms-24-04761-t004:** Cox proportional hazards model with the initial seven variables.

Variable	Beta	HR	HR Lower	HR Upper	z	*p* Value
**age**	**0.059**	**1.061**	**1.038**	**1.084**	**5.404**	**6.52 × 10^−8^**
**sex**	**0.704**	**2.022**	**1.145**	**3.570**	**2.428**	**1.52 × 10^−2^**
LC	0.167	1.181	0.583	2.391	0.463	6.44 × 10^−1^
**log_10_AFP**	**0.650**	**1.916**	**1.477**	**2.486**	**4.892**	**1.00 × 10^−6^**
*HLA-A*33:03*	**0.740**	**2.096**	**1.006**	**4.364**	**1.977**	**4.81 × 10^−2^**
*HLA-A*31:01*	0.277	1.319	0.772	2.253	1.012	3.11 × 10^−1^
*HLA-DPB1*02:01*	−0.256	0.774	0.479	1.252	−1.043	2.97 × 10^−1^

Items with a *p* Value of 0.05 or less are shown in bold.

## Data Availability

The sample and data used for this research were provided from the BioBank Japan Project that was supported by AMED.

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
