# Peer review of "Prediction Model with *HLA-A*33:03* Reveals Number of Days to Develop Liver Cancer from Blood Test"

_ijms, 2023, doi:10.3390/ijms24054761_

Round 1
Reviewer 1 Report
This is a well-written paper, but it needs several improvements.
The introduction.
The link between HLA and HCC in the context of CHB must be clearly stated and referenced.
References.
The references seem to be rather old, which is not criticized. However, there exist a significant body of newer work in the field that is not mentioned.
The body of the article.
The reduction in the number of samples (l. 94-96) is not properly explained. This must be improved.
The use of the BBJ data is not clear in this work.
Selection of the features is described inadequately. It is necessary to justify/explain the decision behind the choices and the methods applied.
A discussion of the features used in the previous approaches with the ones used in the current work should be provided.
Minor points
Only one genotype is used in the regression model. What about the other two?
It is curious to see the instability over time of the sensitivity/specificity presented in Figure 3C. It may be worth bringing up this issue in the discussion.
Author Response
Response to Reviewer #1’s comments
Point 1: The introduction.
The link between HLA and HCC in the context of CHB must be clearly stated and referenced.
Response 1:
GWAS and HLA association analysis which reported that HLA class I genes and HLA class II genes are associated with chronic hepatitis B and liver carcinogenesis are described in the text. Also added the paper to the Reference (reference #8 and #9).
We added and revised sentences as follows.
L87-91,
Genome-wide association studies (GWASs) identified the association of HLA class II genes with chronic hepatitis B infection in Asian populations[8, 9]. As a result of GWAS and HLA association analysis focusing on HCC in Japanese CHB patients, HLA-DPB1*02:01, which belongs to HLA class II, and HLA-A*33:03 and HLA-A*31:01, which belongs to HLA class I, showed a significant association with HCC [10, 11].
Point 2: References.
The references seem to be rather old, which is not criticized. However, there exist a significant body of newer work in the field that is not mentioned.
Response 2:
In addition to the two papers (reference #8 and #9) added in the reply to the comment 1, the following three papers were also added to the reference.
Kim, H. S.; Yu, X.; Kramer, J.; Thrift, A. P.; Richardson, P.; Hsu, Y. C.; Flores, A.; El-Serag, H. B.; Kanwal, F., Comparative performance of risk prediction models for hepatitis B-related hepatocellular carcinoma in the United States. J Hepatol 2022, 76, (2), 294-301.
Yang, H. I.; Yeh, M. L.; Wong, G. L.; Peng, C. Y.; Chen, C. H.; Trinh, H. N.; Cheung, K. S.; Xie, Q.; Su, T. H.; Kozuka, R.; Lee, D. H.; Ogawa, E.; Zhao, C.; Ning, H. B.; Huang, R.; Li, J.; Zhang, J. Q.; Ide, T.; Xing, H.; Iwane, S.; Takahashi, H.; Wong, C.; Wong, C.; Lin, C. H.; Hoang, J.; Le, A.; Henry, L.; Toyoda, H.; Ueno, Y.; Gane, E. J.; Eguchi, Y.; Kurosaki, M.; Wu, C.; Liu, C.; Shang, J.; Furusyo, N.; Enomoto, M.; Kao, J. H.; Yuen, M. F.; Yu, M. L.; Nguyen, M. H., Real-World Effectiveness From the Asia Pacific Rim Liver Consortium for HBV Risk Score for the Prediction of Hepatocellular Carcinoma in Chronic Hepatitis B Patients Treated With Oral Antiviral Therapy. J Infect Dis 2020, 221, (3), 389-399.
Poh, Z.; Shen, L.; Yang, H. I.; Seto, W. K.; Wong, V. W.; Lin, C. Y.; Goh, B. B.; Chang, P. E.; Chan, H. L.; Yuen, M. F.; Chen, C. J.; Tan, C. K., Real-world risk score for hepatocellular carcinoma (RWS-HCC): a clinically practical risk predictor for HCC in chronic hepatitis B. Gut 2016, 65, (5), 887-8.
Point 3: The body of the article.
The reduction in the number of samples (l. 94-96) is not properly explained. This must be improved.
Response 3:
A sentence was added to the text that clinical information on all 672 patients, including missing values, was included in Materials and Methods.
We added the following sentence in the text.
L108-109,
Clinical information of all 672 patients, including missing values, is summarized in Materials and Methods.
Point 4: The use of the BBJ data is not clear in this work.
Response 4:
A sentence stating that variable selection was performed using BBJ samples for the nine items incorporated in the previously reported HCC prediction models is added in the text.
We added the following sentence in the text.
L99-102
Nine test items (including age, sex, HBV-DNA, HBeAg, ALT, LC, PLT, Alb, and AFP) incorporated into five previously reported HCC prediction models (REACH-B, GAG-HCC, PAGE-B, mPAGE-B, and APA-B) were examined to be associated with the development of HCC using BBJ samples.
Point 5: Selection of the features is described inadequately. It is necessary to justify/explain the decision behind the choices and the methods applied.
Response 5:
For the nine items incorporated into the five previously reported HCC prediction models, a sentence was added stating that variable selection was performed on the BBJ sample. In addition, a sentence describing the method of variable selection is included in the text.
We added the following sentence in the text.
L102-103
A stepwise variable selection based on the Akaike information criterion (AIC) was performed for the Cox proportional-hazards regression model with the nine variables.
Point 6: A discussion of the features used in the previous approaches with the ones used in the current work should be provided.
Response 6:
We have added to the Discussion a comparison of the current HCC prediction model with the previously reported models regarding the incorporated items and AUROC for 3-year HCC prediction.
We added the following sentences in the text.
L196-204
Of the four items included in the HCC prediction model constructed in this study, age was included in all five previously reported prediction models. Sex is also an item incorporated into four of the five prediction models (REACH-B, GAG-HCC, PAGE-B, and mPAGE-B). AFP is an item included only in APA-B among the five prediction models, but it is also included in other HCC prediction models, REAL-B[16] and RWS-HCC[17]. An HCC prediction model incorporating the HLA genotype has never been reported before, and the HCC prediction model constructed in this study is the first report. For AUROC for 3-year HCC prediction, the prediction model constructed in this study had AUROC=0.86, which was second only to AUROC=0.88 for GAG-HCC, which had the highest AUROC among the five previously reported prediction models.
Point 7: Minor points
Only one genotype is used in the regression model. What about the other two?
Response 7:
Both HLA-A*31:01 and HLA-DPB1*02:01 were not added as variables because both did not become significant in the variable selection step. We added the result of the variable selection as Table 4.
Point 8: It is curious to see the instability over time of the sensitivity/specificity presented in Figure 3C. It may be worth bringing up this issue in the discussion.
Response 8:
Thank you for the comment. We added to the Discussion that not only AUC but also sensitivity and specificity tend to decrease over time.
We added the following sentence in the text.
L211-213
Although sensitivity at 1 year after blood test fell below 0.7, both sensitivity and specificity tended to decrease over time, as did AUC.
Reviewer 2 Report
The present study is interesting as it focuses on developing a liver carcinogenesis prediction model in Japanese subjects with hepatitis B. The novelty of the model is that it uses parameters such as HLA genotypes for its construction (in addition to those already described in other prediction models such as age, sex, among others). Using this model, it is possible to make a prediction of liver carcinogenesis with the advantage of being able to differentiate between subjects with chronic hepatitis B who developed early from late hepatocellular carcinoma. However, for this manuscript to be considered for publication, the following points must be considered.
1. There is an excessive use of abbreviations in the abstract. They should be described in the text when they are mentioned for the first time, even when they are common like ALT, HBeAg, AUROCs.
2. Keywords cannot be abbreviations.
3. In the results section it is not clear why the BBJ patient base was used for this particular study, perhaps this information should be in the introduction or only in the methodology.
4. Figure 1 is confusing, what is the difference between lines 1 and 4, as well as lines 2 and 5.
5. In figure 2 and 3C, which is presented on the y-axis.
6. Both the introduction and the discussion should be enriched, it should be indicated why the parameters evaluated in this study, such as HLA genotypes, are important. It is also important to mention the limitations of the study.
7. It remains to add the conclusion of the study.
Author Response
Response to Reviewer #2’s comments
Point 1: There is an excessive use of abbreviations in the abstract. They should be described in the text when they are mentioned for the first time, even when they are common like ALT, HBeAg, AUROCs.
Response 1:
Thank you for pointing that out. We have corrected abbreviations in the abstract and text.
Point 2: Keywords cannot be abbreviations.
Response 2:
We have corrected abbreviations in Keywords.
Point 3: In the results section it is not clear why the BBJ patient base was used for this particular study, perhaps this information should be in the introduction or only in the methodology.
Response 3:
A sentence stating that variable selection was performed using BBJ samples for the nine items incorporated in five previously reported HCC prediction models is added in the text. This is the result of selecting variables that are effective in predicting liver carcinogenesis after curation of clinical information of the BBJ patients, so we left the description in Result.
We added the following sentences in the text.
L99-103
Nine test items (including age, sex, HBV-DNA, HBeAg, ALT, LC, PLT, Alb, and AFP) incorporated into five previously reported HCC prediction models (REACH-B, GAG-HCC, PAGE-B, mPAGE-B, and APA-B) were examined to be associated with the development of HCC using BBJ samples. A stepwise variable selection based on the Akaike information criterion (AIC) was performed for the Cox proportional-hazards regression model with the nine variables.
Point 4: Figure 1 is confusing, what is the difference between lines 1and 4, as well as lines 2 and 5.
Response 4:
Figure 1 has been modified to show inclusion/exclusion criteria for the Cox proportional hazards regression analysis. The study design for ROC analysis is summarized in a separate figure (Figure 4).
Point 5: In figure 2 and 3C, which is presented on the y-axis.
Response 5:
The item names of the y-axis are shown in the revised Figure 2 and Figure 3C (Figure 4C in the revised manuscript), respectively.
Point 6: Both the introduction and the discussion should be enriched, it should be indicated why the parameters evaluated in this study, such as HLA genotypes, are important. It is also important to mention the limitations of the study.
Response 6:
GWAS and HLA association analysis which reported that HLA class I genes and HLA class II genes are associated with chronic hepatitis B and liver carcinogenesis are described in the text. Also added the paper to the Reference (reference #8 and #9). We also added the limitation of this study in the Discussion.
We added and revised sentences as follows.
L87-91,
Genome-wide association studies (GWASs) identified the association of HLA class II genes with chronic hepatitis B infection in Asian populations[8, 9]. As a result of GWAS and HLA association analysis focusing on HCC in Japanese CHB patients, HLA-DPB1*02:01, which belongs to HLA class II, and HLA-A*33:03 and HLA-A*31:01, which belongs to HLA class I, showed a significant association with HCC [10, 11].
L221-225
However, because HLA is known to have very different frequencies of genotypes among populations, the limitation of this study is that the HCC prediction model including HLA genotypes constructed in this study may be less effective in non-Asians. In the future, it is necessary to validate the model with prospectively collected cases, and to test the efficacy in non-Asian populations.
Point 7: It remains to add the conclusion of the study.
Response 7:
We added the conclusion of this study at the last of Discussion.
L226-228
The conclusion of this study is that the current model that can predict the development of liver cancer within a year with high accuracy in hepatitis B patients who have specific HLA genotype can be expected to be a clinically useful tool.
Reviewer 3 Report
Review’s comments
The submitted manuscript entitled “Prediction model adding HLA accurately predicts the number of days between blood test and development of liver cancer” focuses on the constructing of a predictive model on the basis of Cox proportional-hazards regression model with four variables including HLA genotypes to distinguish CHB patients with early onset of HCC. This study scientifically sounds and can be of interest for the journal audience. The manuscript contains (17 references total) and about 50% of relevant references published for latest 5 years and five figures and three tables to illustrate the results obtained in the study. However, there are some concerns and recommendations to improve the quality of the manuscript. There are as follows:
1. The title of the manuscript should be corrected. The current title “Predictive… adding HLA accurately predicts…” does not sound good because (i) the phrase “adding HLA” is unclear; and (ii) the authors genotyped the patients for three HLA genes, and (iii) the title contains repeating words.
2. In the Results section, Table 1 shows the results of genotyping of three HLA antigens, including HLA-A*33: 03, HLA-A*31: 01, and HLA-97 DPB1*02: 01. It can be seen from the table that there are no significant differences in prevalence of these HLA types. Therefore, is unclear, what was a reason to choose HLA-A*31:01 to utilize in the prediction model. Consequently, why was not this HLA antigen type indicated in the title of the manuscript?
3. In the results section, more detailed interpretation of all parameters and findings should be provided. For example, what is “beta” in Table 2? How can be interpreted HLA type prevalence values in Table 1? On line 155, the authors stated that the model allowed “detecting patients who would develop HCC within a few years” - what does “the few years” mean? Clarify this.
4. Check the references. For example, see reference No. 16.
5. English language style and grammar should be carefully checked to make the text more readable. Some sentences are not easy to understand.
Author Response
Response to Reviewer #3’s comments
Point 1: The title of the manuscript should be corrected. The current title “Predictive… adding HLA accurately predicts…” does not sound good because (i) the phrase “adding HLA” is unclear; and(ii) the authors genotyped the patients for three HLA genes, and(iii) the title contains repeating words.
Response 1:
Following the comment, we reconsidered the title of the paper and revised it to the following title;
“Prediction model with HLA-A*33:03 reveals number of days to develop liver cancer from blood test”.
Point 2: In the Results section, Table 1 shows the results of genotyping of three HLA antigens, including HLA-A*33:03, HLA-A*31:01, and HLA-DPB1*02:01. It can be seen from the table that there are no significant differences in prevalence of these HLA types. Therefore, is unclear, what was a reason to choose HLA-A*31:01 to utilize in the prediction model. Consequently, why was not this HLA antigen type indicated in the title of the manuscript?
Response 2:
The results of variable selection from seven items, including three HLA genotypes, were summarized in Table 4 and added to the revised paper. One of the three HLA genotypes (HLA-A*33:03) became significant and became one of the items in the current HCC prediction model. We also included this HLA genotype in the title of the paper.
Point 3: In the results section, more detailed interpretation of all parameters and findings should be provided. For example, what is “beta” in Table 2? How can be interpreted HLA type prevalence values in Table 1? On line 155, the authors stated that the model allowed “detecting patients who would develop HCC within a few years” - what does “the few years” mean? Clarify this.
Response 3:
Results of multivariate Cox proportional hazards analysis for seven variables were added to Materials and Methods as Table 4. Each parameter summarized in Table 2 is explained in the text. Because the word in Table 1 is confusing, we have modified the prevalence of each HLA genotype to carrier frequency. Table 3 had the same expression, so we also corrected it. Regarding line 155 of the manuscript before revision, we fixed it within a year because it is important that current HCC prediction model is accurate in predicting HCC within a year.
We added the following sentences to the text.
L122-124
The results of multivariate Cox proportional-hazards regression analysis for seven variables (age, sex, LC, log10AFP, HLA-A*31:01, HLA-A*33:03, and HLA-DPB1*02:01) are described in Materials and Methods.
L128-131
As an example, the beta coefficient for age is 0.057, indicating that older age is associated with a higher risk of liver cancer. Table 2 shows beta coefficient, hazard ratio (HR), confidence intervals of the hazard ratios (HR lower and HR upper), and global statistical significance of the model (z and p.value).
L178-180
The model constructed in this study appears to perform well when detecting patients who would develop HCC within a year, since the mean of Harrell’s C-index and the mean of sensitivity exceeded 0.75 and 0.7, respectively.
Point 4: Check the references. For example, see reference No. 16.
Response 4:
We checked reference No. 16 and didn't find anything strange in our file. The list of reference was stylized using the Endnote software.
Point 5: English language style and grammar should be carefully checked to make the text more readable. Some sentences are not easy to understand.
Response 5:
We reviewed the entire paper and made typographical errors and grammar corrections.
Round 2
Reviewer 2 Report
The authors took in consideration the comments and the manuscript improved and it is clear now. Just a minor comment, the title of tables should be before to show the table.
Author Response
Point 1: The authors took in consideration the comments and the manuscript improved and it is clear now. Just a minor comment, the title of tables should be before to show the table.
Response 1:
Thank you for pointing that out. We have corrected the title of tables.